# Design and Application of Partial Immersion Focused Ultrasonic Transducers for Austenitic Weld Inspection

**DOI:** 10.3390/s22072671

**Published:** 2022-03-30

**Authors:** Yuan Zhang, Zixing Qin, Shizhou Luo, Jeong Hyunjo, Shuzeng Zhang

**Affiliations:** 1School of Traffic & Transportation Engineering, Central South University, Changsha 410075, China; zh_yuan1994@163.com (Y.Z.); 8212190815@csu.edu.cn (Z.Q.); 8212191125@csu.edu.cn (S.L.); 2Department of Mechanical Engineering, Wonkwang University, Iksan 54538, Korea; hjjeong@wku.ac.kr

**Keywords:** austenitic weld, partial immersion focused transducer, B-image, time-dependent threshold

## Abstract

Austenitic stainless steel is a widely used material in the industry, and the welding technique enables stainless steel components to have different shapes for different applications. Any flaws in the weld will degrade the performance of the austenitic component; thus, it is essential to ultrasonically and nondestructively test flaws in welds to ensure service safety. Recently, weld inspection has been performed using contact transducers, but missed detections or false positives for flaws in welds usually occur due to a poor coupling condition in the detection, a low signal-to-noise ratio, and instantaneous noises. In this study, a partial immersion focused (PIF) ultrasonic transducer is designed and used for austenitic weld inspection to address the above issues. The detailed design and manufacture of the PIF transducer are described, and the advantages of the transducer are shown by comparing the results detected using different kinds of transducers. In addition, in order to suppress false positives, a B-image method optimized using a time-dependent threshold is proposed. Experiments are performed to detect flaws in a welded specimen. All the artificial flaws are evaluated using the developed transducer and the proposed method, but minor flaws are mis-detected when planar transducers are used, verifying the method proposed in this paper.

## 1. Introduction

Austenitic stainless steels have been widely applied in pressure vessels, pipelines, and heat exchangers used for petroleum, the chemical industry, nuclear energy, and other industrial fields. Their wide applications are due to their high-temperature fracture toughness, corrosion resistance, creep resistance, and high yield strength in very aggressive environments [1,2,3,4,5]. As a basic manufacturing technology, welding plays an important role in the design and processing of stainless steel components with complicated shapes [6]. However, due to improper processing methods, the improper handling of welding materials, and adverse environmental factors, flaws may occur in welds. In addition, cracks may be produced due to stress, load, fatigue, impact, and irradiation in the service process [7,8,9]. Thus, it is essential to nondestructively detect flaws in welds to avoid accidents caused by structural failure.

Ultrasonic nondestructive techniques are usually used in weld inspection, as they are convenient, safe, and sensitive to potential in-depth flaws [10]. Recently, ultrasonic time-of-flight diffraction (TOFD), ultrasonic phased array, and ultrasonic guided waves have been employed in weld inspection, and detection efficiency, evaluation accuracy, and applicable conditions have been significantly improved [11,12,13,14,15]. However, difficulties still exist in austenitic weld inspection because ultrasonic wave beams are deflected due to anisotropic properties, ultrasonic energy can be decayed due to strong attenuation, and noise and ghost signals can be measured from grain scattering and the reflection of weld boundaries [16]. Missed detections or false positives may occur when flaw signals cannot be extracted from measured signals with a low signal-to-noise ratio (SNR) or when noises and signals reflected from boundaries are treated as flaw signals [17].

It has been proven that the SNR of signals measured from welds can be improved by increasing the acoustic energy density using a focusing technique [18,19]. The focused wave is usually generated by an immersion focused transducer, but in practical application it is impossible to detect flaws in the welds using the immersion method, as the structure can be very large. A focused wave beam in welds can be generated by phased array transducers when contact piezoelectric transducers are used [20]. However, the equipment used in phased array transducers is expensive. To the best of our knowledge, a single contact focused transducer is rarely used, and detection sensitivity for contact transducers is affected by the coupling state. Although an electromagnetic acoustic transducer (EMAT) allows noncontact operation and can generate a focused wave beam by optimizing coils, the influence of noise limits its practical application in weld inspection.

When the weld is scanned using automatic detection, an ultrasonic scan image is usually provided for flaw detection and evaluation. It has been shown that noise information can be obtained from grains or the boundaries of different weld zones [21,22]. It is hard to judge the flaw signal in the image unless one knows of the existing flaws in advance. False positives may occur when an unprofessional or even a professional person evaluates the test results from the image. Thus, when flaws are extracted from the measured signals with a high SNR, filtering noise obtained from the image for flaws will benefit the evaluation process. 

In this study, inspired by the ultrasonic test used for large components in the industry, which involves the use of water spray and water film coupling techniques for objects such as pipelines and steel plates, we propose a ‘quasi-immersion’ ultrasonic test technique using a partial immersion focused (PIF) transducer for weld inspection. A semi-enclosed shell is manufactured, and the element of a focused transducer is immersed in water but the welded structure is not. Ultrasonic waves can be focused in the weld when the PIF transducer is used. The effects of coupling on the wave propagation are minimized, and wave signals with a high SNR can be obtained. A time-dependent threshold method based on statistical noise values is introduced to filter noises and to help extract flaw signals in the imaging process. The design and implementation of the PIF transducer are shown in detail, and a theory for determining the time-dependent threshold is provided. Experiments are performed to test flaws in an austenitic butt weld specimen to verify the ability of the proposed method to reduce missed detections or false positives.

## 2. Materials and Methods

A typical schematic diagram of weld inspection using a single contact transducer is shown in Figure 1a. A longitudinal wave radiated by the transducer propagates in the solid wedge. A shear wave is generated due to wave conversion and propagates in the weld, and the propagating wave is reflected by the flaw and received by the transducer. A typical measured signal is shown in Figure 1b when the contact transducer is used and the system gain is high enough. There are signals reflected by the surface of the specimen, flaws, and weld boundary, and there are noise signals scattered by grains. When the weld is automatically scanned and a B-image is obtained using these measured signals, as shown in Figure 1c, two problems may appear: one is that grain-scattering noises can be treated as flaws (false positives), and the other is that minor flaws may not be found when the amplitudes of the flaw signals are very small (missed detection). Thus, two main approaches are used to solve these problems.

### 2.1. Improve the SNR by Manufacturing Partial Immersion Focused Transducers

As mentioned in the Introduction, focused waves can be introduced to improve the SNR, and this can help to avoid misdetection when flaws are detected in welds and there are significant noise signals. In practical ultrasonic detection, the contact method is usually used, as the welded components are too large to be immersed in water. In this condition, it is not easy to apply a focused transducer in the contact detection method. Thus, a PIF transducer was designed and manufactured to generate a focused wave beam for the weld inspection, as shown in Figure 1d,f. A schematic diagram of the PIF is shown in Figure 2, and the detailed design and implementation will be described in Section 3.

### 2.2. B-Image Method Using a Time-Dependent Threshold

The time-dependent threshold was introduced to filter the noises in the B-image so that the ‘real’ flaws could be clearly shown and the ‘false’ flaws could be excluded. The time-dependent threshold should be obtained based on the statistical results of a large number of reference signals, so a reference weld specimen will be scanned many times and a sufficient number of wave signals will be measured and stored. A mathematical model for determining the time-dependent threshold is described as follows.

First, assume that there are *M* measured wave signals and *N* sampling points in each signal. We use $a_{ij}$ to express the voltage of the *i*-th sampling point in the *j*-th wave signal and V to express the voltage amplitude set, which is described as:(1)$$V=\left(\begin{array}{l} a_{11}\cdots a_{i1}\\ \vdots \;\;\;\ddots \;\;\;\;\vdots \\ a_{1j}\cdots a_{ij} \end{array}\right),a_{ij}\in V,i=1,2,\ldots N;j=1,2,\ldots M$$

Second, the maximum and minimum voltage values at each sampling point from all the measured signals are extracted and stored in matrices, and *T* segments are set using the minimum and maximum values as the lower and upper limits, respectively. The number of segments is determined considering the minimum and maximum values, and it is proven that good results can be obtained when 50 segments are used. Take the results for the first sampling point values as an example. The maximum and minimum values are extracted as:(2)$$a_{1,\max}=\max(a_{1,j});a_{1,\min}=\min(a_{1,j})\;\;j=1,2,3,\ldots N$$
and the number of values in the *k*-th segment is counted as:(3)$$N\left(k\right)=count\left(a_{1,\min}+(k-1)\frac{a_{1,\max}-a_{1,\min}}{T}\leq a_{1,j}<a_{1,\min}+k\frac{a_{1,\max}-a_{1,\min}}{T}\right),k=1,2,\ldots T$$

Based on the statistical results obtained for noise and boundary-reflected signals, we can select a voltage amplitude *A*_1_ as the threshold when 99% of the signal amplitudes are less than this value or use a larger amplitude to remove more noise. Note that $a_{1,\max}$ is usually not treated as the threshold, as there are several abnormally large noises.

Last, when the thresholds for all sampling points are obtained, all these values are combined to form a new signal, G, which is the required time-dependent threshold.
(4)$$G=\left(A_{1},A_{2},\ldots A_{N}\right)$$

Note that there should be no flaws in the reference weld specimen, and the reference weld specimen should preferably be the same as the weld specimens to be examined. When the scanning experiment is performed and reference signals are measured, the distance between the PIF transducer and the weld is made as large as possible. The detailed theory used to determine the time-dependent threshold can be found in published works [21,23].

A demonstration diagram of the time-dependent threshold is shown in Figure 1g. When the B-image is obtained, only the signals whose amplitudes are higher than the time-dependent threshold will be treated as flaw signals and be shown; thus, noises or false flaws can be removed, and only flaws will remain, as shown in Figure 1h.

To summarize the method in one sentence, a PIF transducer is designed for the weld inspection (Figure 1d) to improve the SNR (Figure 1e), and a time-dependent threshold is introduced to filter noise and determine the presence of flaws (Figure 1f); thus, a B-image detection result with a high accuracy can be obtained.

## 3. Design and Implementation of PIF Transducers

In the design of the PIF transducer, water was selected as the first medium, and the longitudinal wave radiated by the transducer propagated in this medium. In the weld inspection, the shear waves were used as they are more sensitive to the defects that are harmful to the weld, such as cracks, incomplete penetration, and incomplete fusion. Thus, oblique incidence and mode conversion were introduced to ensure shear waves propagate in the weld specimen. The incident angle can be selected using Snell’s law, which is expressed as:(5)$$\frac{c_{p1}}{\sin\alpha_{}}=\frac{c_{p2}}{\sin\beta_{}}=\frac{c_{s2}}{\sin\lambda }$$
where $\alpha_{}$ is the incidence angle and $c_{p1}$ is the longitudinal wave in the first medium; β and λ are refraction angles of the longitudinal and shear waves, respectively; and $c_{p2}$ and $c_{s2}$ are the longitudinal and shear waves, respectively, in the second medium.

The speed of the longitudinal wave in water is about 1480 m/s, and the speeds of the longitudinal and shear waves in the weld specimen are about 5970 m/s and 3280 m/s, respectively. We can obtain the first and second critical angles, which are 14.3° and 27°, respectively. In this study, the refraction angle for the shear wave is set at 49° to ensure that most of the weld zone can be covered by the wave beam. In this condition, the incidence angle is about 20°. Thus, the incidence angle is located between the first and second critical angles, and the effects of the refracted longitudinal wave on the measurement can be minimized.

One advantage of the PIF transducer is that the focused transducer can be used to detect the weld, which can improve the SNR. The relationship between the geometrical focal length, $f_{d}$, and the propagating distance of the shear wave in the weld specimen, $d_{2}$, can be described as:(6)$$f_{d}=d_{1}+\frac{c_{s2}}{c_{p1}}d_{2}$$
where $d_{1}$ is the wave propagating distance in water. In the detection, according to empiricism, the focused point is selected at the middle of the specimen. When the calibrated $f_{d}$ of the transducer is 48 mm, the thickness of the weld specimen is 16 mm and the refraction angle is 49°. We determine that $d_{1}$ is 26 mm.

The shell of the PIF transducer is designed based on the selected parameters above and the shape of the transducer. The shell was designed based on the manufacturing method of the shell, the fixation of the transducer, the water-inlet mode and water-saving mode, and the clamping of the PIF transducer on the mechanical detection frame, and its schematic diagram is shown in Figure 3. The ultrasonic immersion transducer was placed in the inclined hole in the raised component of the shell, and a screw was used to fix the transducer to ensure the incident angle and underwater acoustic distance. There were two impermeable holes on both sides of the raised component connected to the manipulator. There was a pipe in the upper part of the shell that was connected to the water pipeline for providing coupling water. The interior of the shell was hollow, and the bottom was open. The bottom of the shell was made of a soft rubber material. The silica rubber is soft and flexible so that it can attach to the surface of the weld seamlessly to ensure a good coupling state and save water. The front part of the shell was designed as a semicircle, and the internal wall was set as a sawtooth to increase the wave diffuse reflection and reduce the influence of clutter wave signals.

The geometry of the manufactured PIF transducer and the manipulator located on the weld specimen is shown in Figure 4. The shell of the PIF transducer was manufactured using 3D printing technology. The bottom material was silica gel, and the rest was polylactic acid [24]. Self-tapping screws were used for reaming the threaded hole in the raised component so that a plastic screw could be used to fix the transducer more stably. The water inlet of the shell was connected with a rubber pipe, and the water could then flow into the shell due to gravity. When the silica gel of the shell came into contact with the specimen, a good sealing performance could be achieved, meaning that water could fill the shell and be retained there during the detection process.

## 4. Experiments

### 4.1. Specimen Preparation

Two identical austenitic stainless steel butt welds were designed and manufactured. One was used to analyze wave signals for comparison and the weld was well made and there were no flaws inside. There was a crack, an incomplete penetration defect, and a manufactured flat bottom hole in the other specimen. The schematic diagram, picture, and Radiographic Testing (RT) scanning image of the specimen with flaws are shown in Figure 5a–c, respectively.

Before welding, the surfaces of the austenitic stainless steel plates were polished to remove rust stains and V-grooves with an angle of 70° were machined at the welding position of the two plates. The two plates were welded using the shielded metal arc welding method. When the internal incomplete penetration flaw was created, a thicker electrode and a lower current during welding were used. When the internal crack was created, the weld of the specimen, whose length was slightly longer than the length of the designed crack, was welded locally first. Then, the weld was broken with an external force and the broken part was sealed with a low welding current. The flat bottom hold, whose diameter was 0.3 mm and whose depth was 0.3 mm, was manufactured by machining with a special drill bit. Detailed information on the position of the defects can be found in Figure 5a.

### 4.2. Experimental Process

Experiments were performed to scan the weld specimen using the designed PIF transducer and the detection system, as shown in Figure 6. The pulse/echo measurement was conducted using a pulser/receiver (JSR DPR-300), a PIF transducer (transducer: Olympus V306, focal length: 2 inch; central frequency: 2.25 MHz; diameter: 0.5 inch), and a DAQ card (AD Link PCIe-9852). The PIF transducer was controlled by a mechanical inspection frame, and an encoder was used to record the moving distance and the trigger signal. The waveforms were collected at a sampling rate of 200 MHz. As the weld had a higher acoustic impedance, the energy of the transmitted shear wave was very small. Therefore, the system gain of 45 dB was selected to ensure that the shear wave in the weld could be measured.

Three other experiments were also performed. In the first one, the reference specimen was scanned using the same PIF transducer to analyze the wave signal and determine the time-dependent threshold. In the second and third one, the specimen with flaws was scanned using a partial immersion planar (PIP) transducer and a contact shear wave transducer, respectively, for comparison.

## 5. Results and Discussions

### 5.1. Typical Wave Signal Analysis

Typical A-wave signals measured using the PIF transducer are shown in Figure 7. The reference signal obtained from the reference weld specimen is shown in Figure 7a. Noise signals and signals reflected by the interface and boundaries of the weld zone can be observed. When the weld specimen with flaws was detected and the wave beam was reflected by the flat-bottom hole (FBH), the wave signal was measured, as shown in Figure 7b. Compared with the reference signal shown in Figure 7a, the flaw signal with a high SNR could be distinguished as the focused wave beam was generated. Good coupling conditions were achieved, as the wave signals reflected by the flaw and the boundary of the weld were measured well.

### 5.2. Traditional B-Scan Image

When the weld specimen with flaws was scanned, a B-image was obtained using the A-wave signals measured by the PIF transducer, as shown in Figure 8. The flaws from incomplete penetration, cracks, and FBH are marked in the B-image. However, there were some suspected flaw signals outside of the marked regions in the image, which would influence the judgment of the defects. Thus, an optimized B-image obtained using the time-dependent threshold was used to differentiate the false positive flaw detection. 

### 5.3. Optimized B-Scan Image

The time-dependent threshold was determined in advance in order to filter suspected flaw signals in the B-image. Based on the theory and implementation process described in Section 2.2, the reference specimen was scanned and 10,000 A-wave signals were measured and stored in order to extract the signal extremum profile. The statistical results obtained at the sampling times of 22 and 29 μs are shown in Figure 9. In the statistical analysis, 50 segments were used at each sampling position. In the scanning process, the distances between the transducer and the weld may slightly change, the shapes of the weld components in different positions may be somewhat different, the surface roughness may affect the wave reflection and propagation, and there may be several signals with abnormally large local signals due to the randomness of the grain noise. Therefore, 99.5% of the statistical wave voltage amplitude at each sampling point was selected as the signal extremum profile. The determined time-dependent threshold and some wave signals are shown in Figure 10. It was determined that the reference wave signals were within the threshold and parts of the flaw signals were out of the threshold.

When the time-dependent threshold is introduced to extract the flaw information, the optimized B-image is obtained, as shown in Figure 11. Compared with Figure 8, all the identified flaws remain but the ultrasonic spike from the flaw is reduced. When the optimized B-image is used, it is more conducive to testing and evaluating the flaws in the welds. In the experiment, a flaw whose equivalent size is about 1/4 of the wavelength is detectible, as the shear wave is sensitive to volumetric defects and the time-dependent threshold can be used to filter the noise. Note that when the B-image is used and there are abnormal signals in the position where the flaw appears, one can only judge whether there is a flaw in this position, but it is difficult to judge how many flaws there are in this study. Additionally, note that not all noises can be filtered, as there will be abnormal signals that appear, but these components can be easily distinguished from the flaw signals because they show speckle-like distributions. Our repeated experimental results show that the properties of the abnormal signals are unstable, with their numbers and positions changing in different experiments, although these results are not shown in this study.

### 5.4. Comparison of Results Determined Using Other Detection Transducers

In order to show the advantages of the developed PIF transducer in the weld inspection, the weld specimen with flaws was scanned using a partial immersion planar (PIP) transducer (transducer: Olympus, I3-0208-S; central frequency: 2.25 MHz; diameter: 0.5 inch) and a planar contact shear wave transducer (transducer: Olympus; V404, central frequency: 2.25 MHz; diameter: 0.5 inch). When the PIP transducer was used, the experiment was performed with the same experimental setup as described in Section 4.2. Some A-wave signals are shown in Figure 12a, and the corresponding B-image is shown in Figure 12c. It was determined that the flaws of incomplete penetration and cracks can be found in the original B-image because these flaws were very large. However, there was little detection information for the FBH in the B-image, and missed detections often occurred. The comparison results of the A-wave signals in Figure 12a also indicate that the flaw signal was submerged in the noise signal.

When the planar contact shear wave transducer was used, the experimental system gain was set at 40 dB. The A-wave signals measured for the FBH and the normal weld part are shown in Figure 12b, and the corresponding B-image is shown in Figure 12d. It is shown that the A-wave signal determined by the contact transducer was very different from that obtained using the immersion transducer, and its main property was that the signal component reflected by the boundary of the weld had the maximum amplitude. Similar to the results determined by the PIP transducer, only the flaws of incomplete penetration and cracks could be detected. In addition, when the contact transducer was used, it was difficult to guarantee a good coupling condition; thus, the wave signals were not stable, as shown by the deformed parts of the B-image.

Note that as the FBH was not found in the B-image when the contact transducer and the PIP transducer were used, the B-image was not optimized using the time-dependent threshold. The comparison results obtained using the developed PIF transducer and other transducers indicated that the detection accuracy and resolution are better when the proposed method was introduced, as both missed detections and false positives for flaws were reduced.

## 6. Conclusions

In this study, a B-imaging method for austenitic weld inspection based on using a designed PIF transducer was proposed for the experiments and a time-dependent threshold was proposed for filtering noise. The focused wave beam was generated by the PIF transducer and radiated into the weld specimen to improve the SNR. A good coupling condition was also achieved, as water could remain between the transducer and the specimen. A time-dependent threshold was determined by analyzing the extreme value distributions of the reference signals and introduced to filter the noise in the B-image. The experiment was performed using the designed PIF transducer and the B-image was optimized using the time-dependent threshold. It is shown that the manufactured flaws in the weld specimen were all accurately detected using the proposed method. Comparison experiments were performed using a PIP transducer and a traditional planar contact transducer. It was determined that missed detection happens when the flaw has a submillimeter size.

Applying the PIF transducer for weld inspection in laboratory conditions was a meaningful experiment. A very different. It may be challenging to obtain accurate statistical results for noise. There are also various types of welded joints. In future studies, a grain backscattering model for welds will be researched to theoretically predict the distributions of noises in order to obtain the time-dependent threshold.

## Figures and Tables

**Figure 1 sensors-22-02671-f001:**
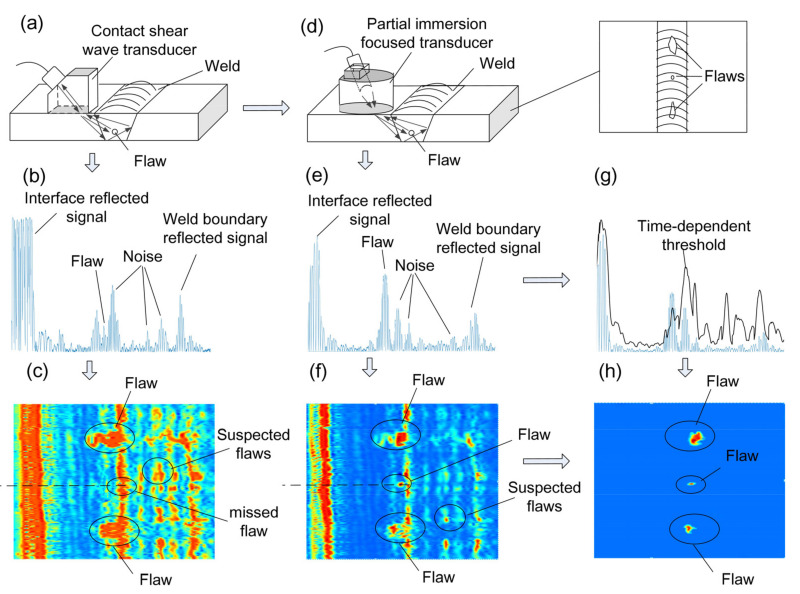
Schematic diagram of weld inspection. (**a**) traditional contact ultrasonic detection, (**b**) typical ultrasonic wave signal, (**c**) imaging result using contact detection method, (**d**) proposed partial immersion focused ultrasonic detection, (**e**) enhanced ultrasonic wave signal, (**f**) imaging result using partial immersion detection method, (**g**) time-dependent threshold for the ultrasonic wave signal, and (**h**) optimized imaging result for flaws.

**Figure 2 sensors-22-02671-f002:**
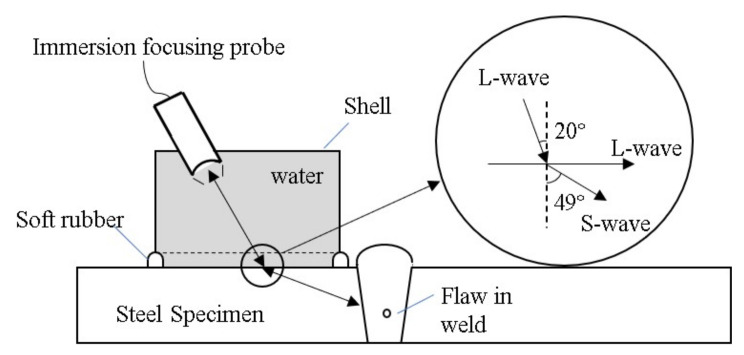
Schematic diagram of the designed PIF transducer.

**Figure 3 sensors-22-02671-f003:**
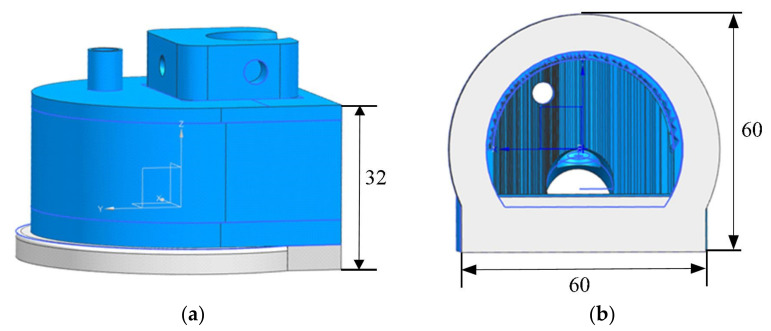
Designed shell from (**a**) the side view and (**b**) the bottom view.

**Figure 4 sensors-22-02671-f004:**
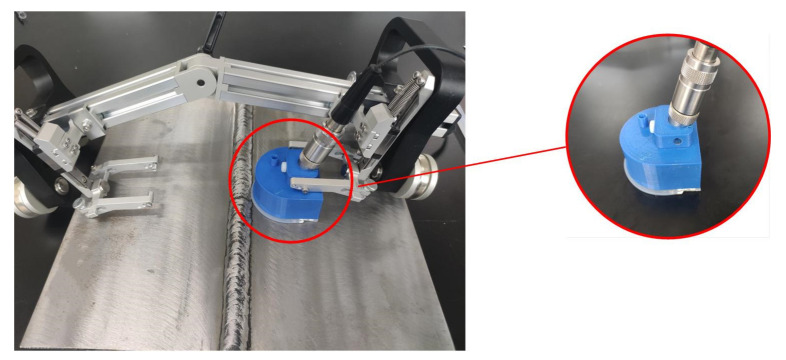
Manufactured PIF transducer and manipulator.

**Figure 5 sensors-22-02671-f005:**
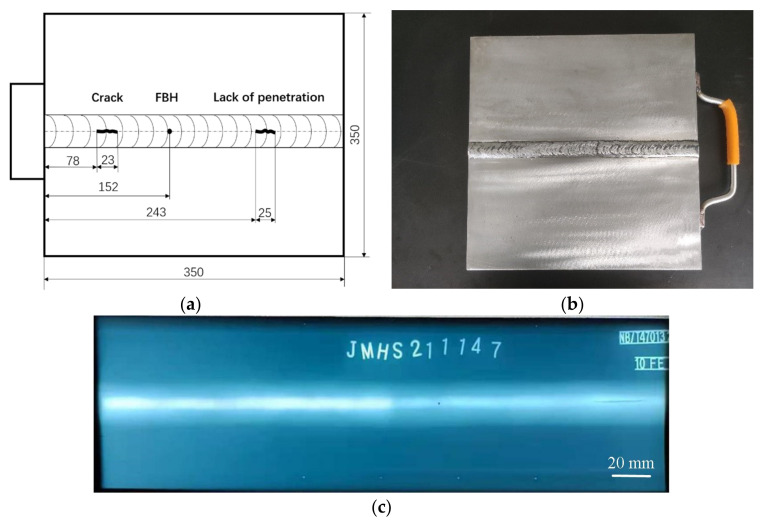
(**a**) Schematic diagram, (**b**) physical picture, and (**c**) RT scanning image of the weld specimen with defects. Note that in (**c**), the crack was not found using the RT method, as it was an area-type flaw.

**Figure 6 sensors-22-02671-f006:**
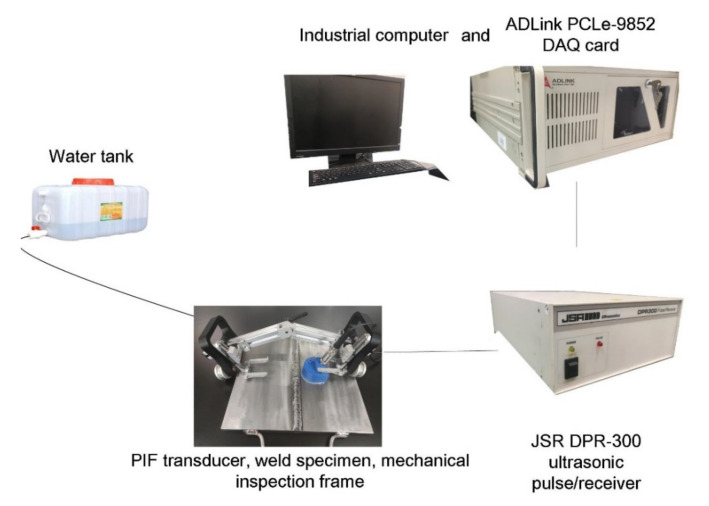
Experimental setup.

**Figure 7 sensors-22-02671-f007:**
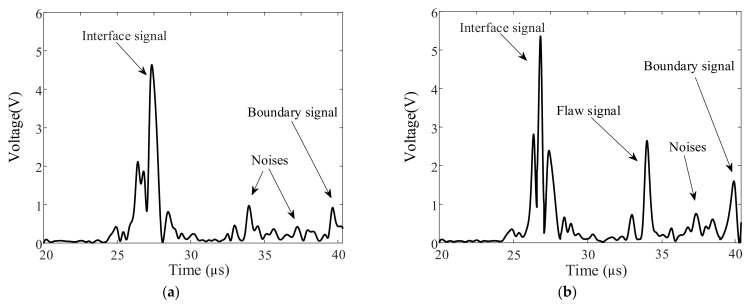
Typical A-wave signals measured using the PIF transducer for (**a**) the reference weld component and (**b**) the weld with a flaw.

**Figure 8 sensors-22-02671-f008:**
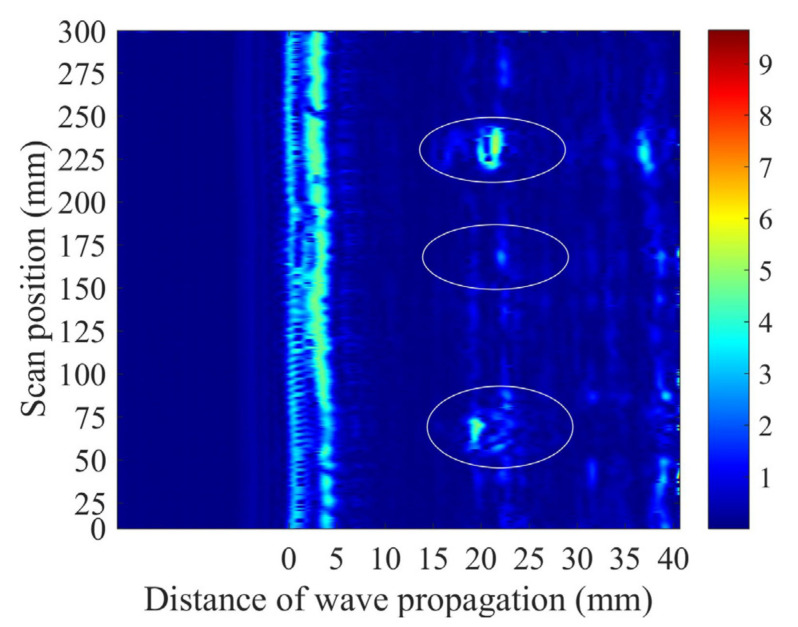
B-image for the weld specimen obtained using the A-wave signals. The real flaws and their positions are marked in the ellipses, but there are suspected flaws outside these regions.

**Figure 9 sensors-22-02671-f009:**
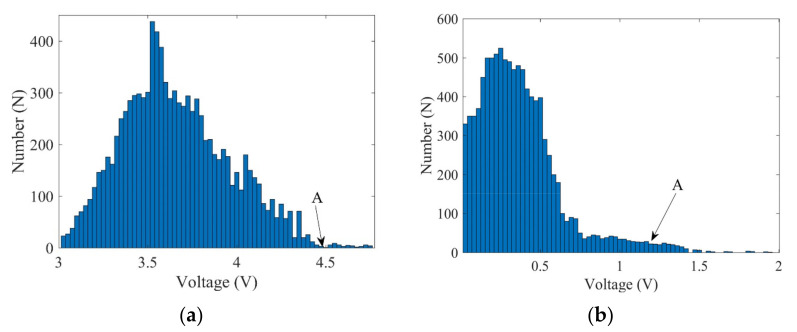
Statistical results obtained for wave voltage amplitudes at (**a**) 22 μs and (**b**) 29 μs.

**Figure 10 sensors-22-02671-f010:**
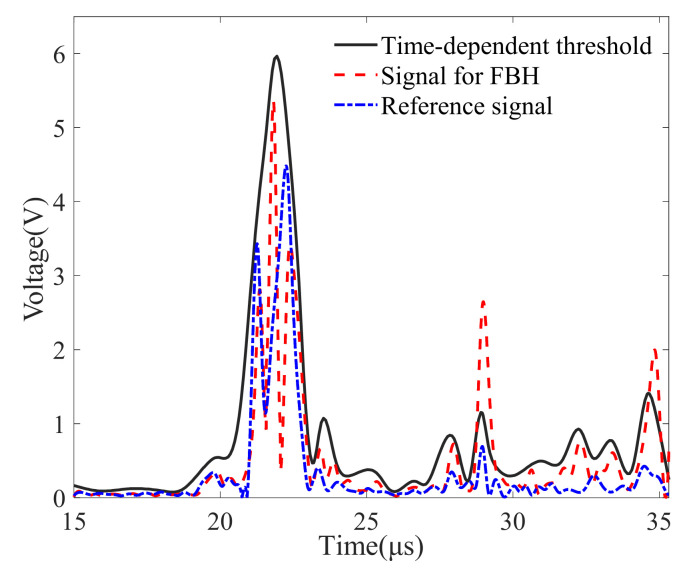
A-wave signals and time-dependent threshold.

**Figure 11 sensors-22-02671-f011:**
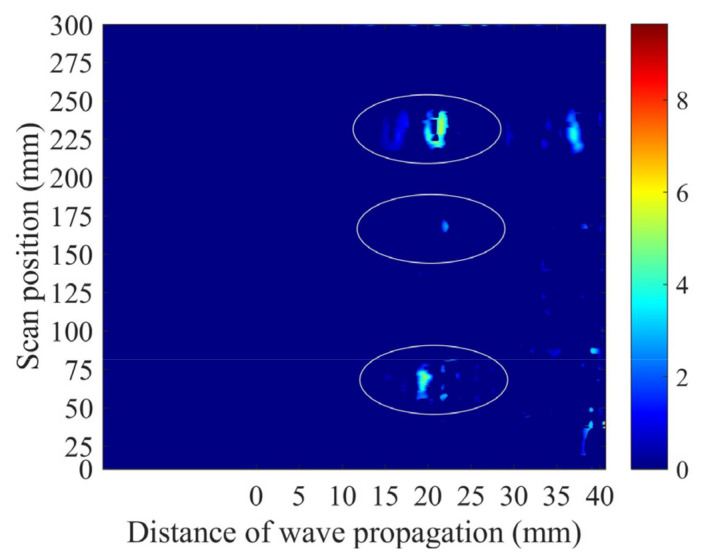
B-image for the weld specimen obtained using the A-wave signals and optimized using the time-dependent threshold. The real flaws and their positions are marked in the ellipses.

**Figure 12 sensors-22-02671-f012:**
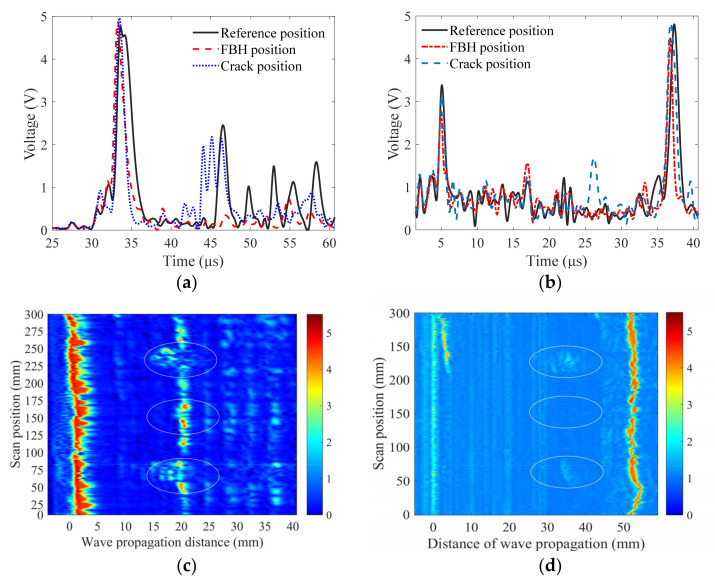
Weld inspection results. (**a**) A-wave signals and (**c**) B-image determined using a PIP transducer. (**b**) A-wave signals and (**d**) B-image determined using a contact shear wave transducer. The real flaws and their positions are marked in the ellipses, but in (**c**) there are many suspected flaws with very high amplitudes near the FBH, and in (**d**) the FBH flaw is mis-detected.

## Data Availability

The data that support this study are available from the corresponding author upon reasonable request.

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
