# Peer review of "Design and Application of Partial Immersion Focused Ultrasonic Transducers for Austenitic Weld Inspection"

_sensors, 2022, doi:10.3390/s22072671_

Round 1

Reviewer 1 Report

The authors present the applications of PIF for flaw detection. Overall, this work has the innovation of using silica shell to hold to water and avoid large amount of water as the coupling medium. However, there are some issues that needs to be addressed:

  1. Missing unit in the figures, for example, what is the colorbar of the Figure 1 (c). Missing scale, for example, figure 2. Figure 1 (a, d, g) too small and hard to be identified.
  2. For shear wave method, how the longitudinal wave convert to the shear wave? Since the weld has high acoustic impedance, the longitudinal wave is more likely to reflect back, please explain.
  3. It will be better if a schematic diagram of the PIF is presented in detail, this diagram will help readers to better understand. Also, please address in the manuscript that the silica rubber is soft and flexible, so that it can attach the surface of the weld seamlessly.
  4. Delete Line 143-145.
  5. A discussion part should be provided about the minimum crack can be detected in theory, and how likely to differentiate the false positive crack detection.

Reviewer 3 Report

The submitted manuscript proposes a novel NDT method that uses a focused probe by partially immersing the transducer and inspection surface in water, and B-image method using a time-dependent threshold, both of which improve detection accuracy. By comparing the B-imaging of a partial immersion planar transducer and a planar contact shear wave transducer, the advantages of this method for the detection of small defects are verified. However, defect detection in water is a common method, and the advantages of the proposed device need to be demonstrated.

  1. NDT is often carried out in water. Using a focused transducer in water makes the device simpler. Are there any advantages to the method described in the article?
  2. What are the advantages of the device described in comparison to the potable crystal probe that also has a focusing function?
  3. In line 119, the value of k changes from 1 to T, how to determine the value of the upper limit T, please explain.
  4. In Figures 7,10, 11(c) and 11(d), you may need to mark the names and locations of the flaws.
  5. In Figure 2(b), the designed shell was not clearly present. Please add detailed information.

Round 2

Reviewer 2 Report

The article was improved. All comments made by the reviewer were addressed and most resolved satisfactorily. I consider that the article is of interest to other researchers and engineers and has the quality required for its publication.